# A mechanochromic donor-acceptor torsional spring

Maximilian Raisch [ID] [1], Wafa Maftuhin[2,3], Michael Walter [ID] [2,3,4 ✉] & Michael Sommer [ID] [1 ✉]

Mechanochromic polymers are intriguing materials that allow to sense force of specimens under load. Most mechanochromic systems rely on covalent bond scission and hence are two-state systems with optically distinct "on" and "off" states where correlating force with wavelength is usually not possible. Translating force of different magnitude with gradually different wavelength of absorption or emission would open up new possibilities to map and understand force distributions in polymeric materials. Here, we present a mechanochromic donor-acceptor (DA) torsional spring that undergoes force-induced planarization during uniaxial elongation leading to red-shifted absorption and emission spectra. The DA spring is based on *ortho*-substituted diketopyrrolopyrrole (*o*-DPP). Covalent incorporation of *o*-DPP into a rigid yet ductile polyphenylene matrix allows to transduce sufficiently large stress to the DA spring. The mechanically induced deflection from equilibrium geometry of the DA spring is theoretically predicted, in agreement with experiments, and is fully reversible upon stress release.

[1] Institut für Chemie, Technische Universität Chemnitz, Chemnitz, Germany. [2] FIT Freiburg Centre for Interactive Materials and Bioinspired Technologies, University of Freiburg, Freiburg, Germany. [3] Cluster of Excellence livMatS @ FIT, Freiburg, Germany. [4] Fraunhofer IWM, MikroTribologie Centrum µTC, Freiburg, Germany. ✉email: Michael.Walter@fmf.uni-freiburg.de; michael.sommer@chemie.tu-chemnitz.de

Mechanochromic polymers are able to change their color of absorption or emission as a response to mechanical force. This visualization of molecular deformation[1–4] and covalent bond cleavage[5–9] sheds light on the force transduction at the single polymer chain level and offers ample possibility for applications, such as force sensor design, tamper-proof packing, and safety of materials[10]. Likewise, mechanochromic materials are in high demand for the investigation of fundamental physical and biological questions, including visualization of force distributions in Gecko-type adhesives[11] or cell hydrogel interactions relevant to tissue engineering and cell differentiation[12]. Mechanochromic materials are known for some time[1,3], but activity in this area is increasing since the last decade after seminal works on force-induced C–O bond cleavage of spiropyran (SP) furnishing colored merocyanines (MC), which has been demonstrated in solution[13] and in bulk materials[14]. Such two-state systems represent a mechanically triggered optical switch with "on" (MC) and "off" (SP) states. Other mechanophores such as Diels-Alder adducts[15], various spiro compounds[7,16,17], or precursors for stable radicals[5,18,19] are based on this general principle as well. Mechanistically different is mechanically induced cleavage of dioxetane-containing polymers leading to excited states, which undergo energy transfer to chromophores and light emission in the visible region[20]. Next to covalent systems, supramolecular interactions enable the design of mechanoresponsive materials, using processes that rely on distinct optical properties of differently stacked chromophores[1,2,21].

Depending on application and mode of detection, reversibility of mechanophores may or may not be required. Most of the mentioned mechanophores make use of a mechanically triggered chemical reaction and thus involve a two-state system in the most simple case. Here, one advantage lies in the tunable but typically long lifetime of the mechanically induced "on state" after stress release, which is critical for signaling damage in e.g., glass fiber composites[22] once exposed to a critical overload. After stress release, a fast back reaction to the initial, active state often demands external energy like heat[21] or UV-light[6,17]. In the case of SP, the barrier of the back reaction depends sensitively on the substitution pattern and can be lowered to obtain a transient mechanochromic response that is only stable under-maintained stress[23]. Yet, in order to allow for the investigation of dynamic mechanical processes or online monitoring of stress distributions, two-state systems are rather inappropriate. Another drawback of two-state systems is their lack of differentiation between different forces, as soon as the activation energy is reached, i.e., the restriction to a switch between "on" and "off". An interesting but sophisticated remedy to this restriction is the usage of multiple mechanophores in polymer blends[24] or composites[25].

Another principle of mechanochromism in polymers that has received far less attention relies on conformational changes in conjugated π-systems such as polydiacetylenes[3,4] or poly(3-alkylthiophene)[26,27]. Reported examples of force-induced conformational changes in conjugated polymers include polyfluorenes[28,29] and polythiophenes[30,31] where coplanarity is possible due to rather shallow torsion potentials. In most cases[28,29,31], blending approaches were used to attenuate absorption of specimens under scrutiny, which is associated with morphological and processing issues, and eventually limits the scope of applications.

Here, we report a donor (D)-acceptor (A) mechanochromic torsional spring that exhibits a large dihedral angle in the equilibrium geometry. The mechanochromic response originates from the force-induced planarization of D and A, which leads to increased electronic communication and hence marked changes in the optical properties. Our system is based on differently substituted diphenyldiketopyrrolopyrroles (PhDPP)

that are covalently incorporated into the backbone of an amorphous, rigid yet tough polyphenylene. As the mechanically induced planarization corresponds to a deflection from equilibrium geometry, stress release gives rise to fast recovery of the initial optical properties, despite the rigid polyphenylene matrix. Consequently, the bathochromic shift of both absorption and emission bands is directly related to the tensile stress acting on the polymer chain. These results are corroborated by detailed theoretical calculations, including tensile testing, which confirms force-induced planarization and the associated optical changes. The calculations furthermore reveal qualitative forces acting on the PhDPP units.

## Results

**Design of a DA torsional spring**. The principle of the herein presented mechanochromic torsional spring is based on a pair of an electron-rich donor (D) and an electron-deficient acceptor (A) brought into conjugation. Sterical hindrance between D and A results in a significant deviation from coplanarity at equilibrium geometry that only allows for partial conjugation of the π-systems (Fig. 1a). This situation causes both, a hypsochromic shift (a shift to smaller wavelengths) of the charge-transfer (CT) absorption band in the UV-vis spectrum and a decrease in the oscillator strength compared to the fully coplanar DA-system. If the position of polymer chain attachment at the DA spring is appropriately chosen, elongating the end-to-end distance of the chain by macroscopic force may lead to planarization of the DA spring (Fig. 1b). The resulting increase in conjugation leads to a bathochromically shifted CT band and an increase in oscillator strength (Fig. 1c). Consequently, a release of the external force restores the initial conditions with respect to the geometry and optical properties of the DA spring.

As a DA system, a DPP core (A) with two *ortho*-substituted phenyl rings (D) as flanking units was chosen. N-methylation of the lactam rings impedes intermolecular H-bonding and strong aggregation between DPP chromophores (Fig. 1d)[32]. To evaluate the suitability of the *ortho*-substituted PhDPP (*o*-DPP) as DA spring, we start from bromine-substituted diphenyl DPP. *Ortho*-substitution of the phenyl ring with bromine (*o*-BrPhDPP) increases the torsional angle compared to more often used *para*-Br functionalization including *ortho*-hydrogens (*p*-BrPhDPP). While our DFT calculations reveal a dihedral angle of 30° for *p*-BrPhDPP similar to other *p*-PhDPPs[33,34], sterical hindrance enforces a more than doubled dihedral angle of 67° for *o*-BrPhDPP (Supplementary Fig. 1, Fig. 2a).

To further evaluate the suitability of the *ortho*-substituted PhDPP as DA spring, we calculated how the rotation of the BrPh-group with respect to the DPP core changes the energy and optical properties of BrPhDPP (Fig. 2b–d). The corresponding torsion potentials for the simultaneous rotation of both flanking BrPh groups in *p*-BrPhDPP and *o*-BrPhDPP are depicted in Fig. 2b (very similar potentials are obtained when rotating single BrPh groups in hypothetically asymmetric DPP molecules, see Supplementary Fig. 2). Moderate sterical hindrance drives the equilibrium configuration of *p*-BrPhDPP away from the electronically preferred coplanar conformation. This effect is much stronger for *o*-BrPhDPP, where the equilibrium dihedral angle is much larger, as well as the maximum energy of up to 200 kJ mol$^{-1}$ required to reach full coplanarity. The observed asymmetry in the torsion potentials is caused by the different interactions of the BrPh ring with the methyl and carbonyl groups of the DPP core. Full rotation, therefore, shows a hysteresis that effectively stabilizes the orientation to one side (Fig. 2b).

Figure 2c shows the experimental and calculated UV-vis spectra of *o*-BrPhDPP and *p*-BrPhDPP. The spectra are in good qualitative

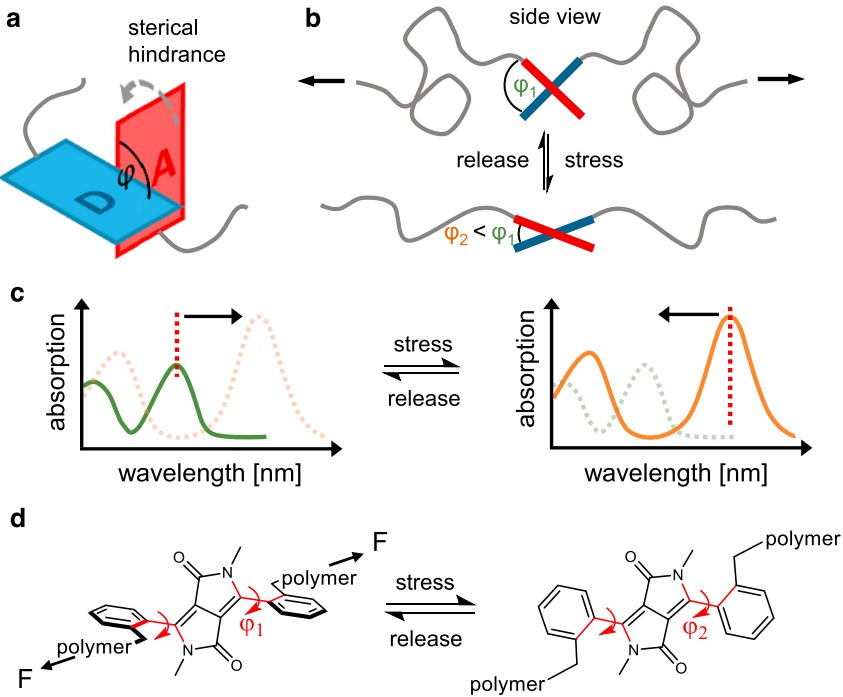

**Fig. 1 Principle of a mechanochromic torsional spring. a** A donor–acceptor (DA) unit with a large torsional angle φ is covalently incorporated into a polymer chain. **b** Stretching of the polymer backbone causes a decrease of φ. **c** Schematic reversible shift of the UV-vis-absorption band as a response to elongational stress. **d** Structure of the DA-system reported here based on diphenyldiketopyrrolopyrrole (PhDPP).

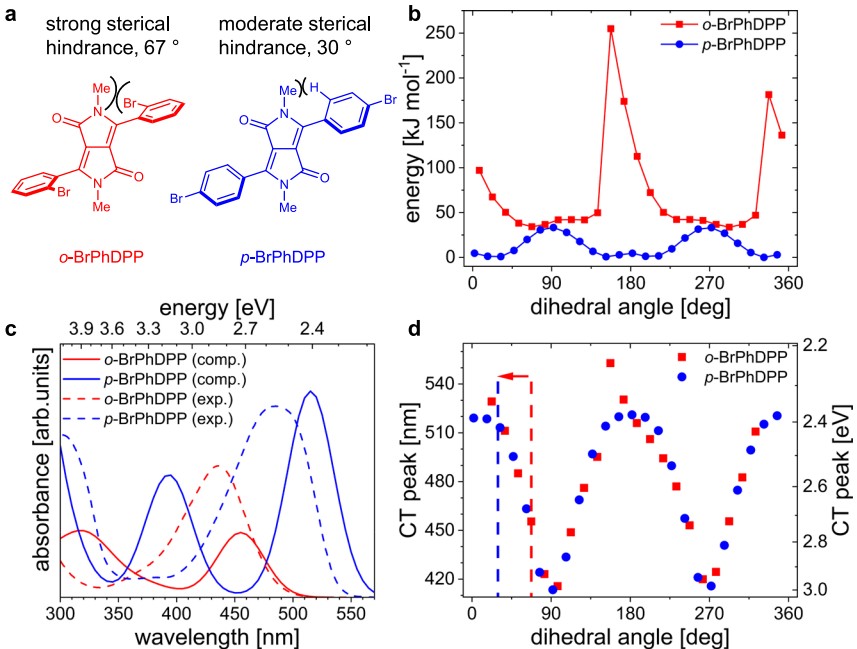

**Fig. 2 Theoretical evaluation of *ortho*-substitution in PhDPP. a** Molecular structures of *o*-BrPhDPP (red) and *p*-BrPhDPP (blue). **b** Computed torsion potentials for the rotation of the BrPh group. **c** Simulated UV-vis spectra for the relaxed structures in the gas phase and experimental UV-vis spectra in DMF (0.035 mmol L$^{-1}$). **d** Variation of the simulated CT-peak position with the dihedral angle. Dashed lines indicate equilibrium geometries (minima in the torsion potentials in **b**), and the arrow the direction if an external force is applied.

agreement, in particular for the relative shift of the low energy bands of *o*-BrPhDPP and *p*-BrPhDPP. In theory and experiment, these bands differ by 0.31 eV and 0.30 eV, respectively. The possibility to detect geometric changes by optical changes (or vice versa) can now be inferred from Fig. 2d, which displays the change of the absorption wavelength of the CT band with varying dihedral angles. Deviations of 15° and 30° from the equilibrium geometry shift the CT band by 30 nm and 55 nm, respectively. This estimation confirms the diphenyl DPP system as a promising candidate for an experimental realization of a DA spring.

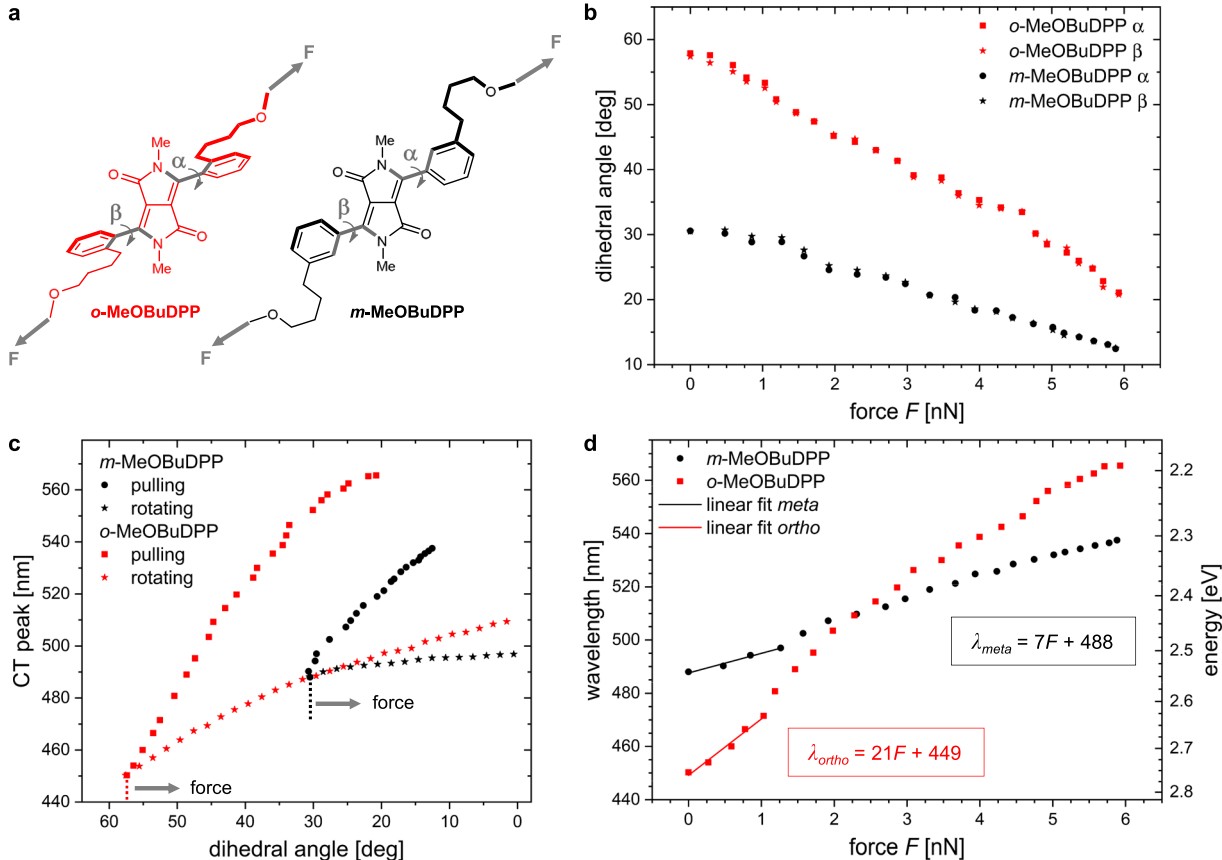

**Fig. 3 Theoretical elongation of the DA spring. a** Structures of the *ortho*-4-methoxybutyl and *meta*-4-methoxybutyl functionalized DA spring (MeOBuDPP) for theoretical investigations. **b** Calculated relation between external elongational force (*F*) and the dihedral angles α, β between Ph and DPP core. **c** Analysis of the contributions from changes in the dihedral angles and deformation of the structure on the optical properties. **d** Calculated relation between *F* and both wavelength and energy of the charge-transfer absorption band of *o-/m*-MeOBuDPP.

**Theoretical investigation of the DA spring under force**. We now turn to the question of the effect of force acting on the DPP spring once covalently incorporated into a polymer matrix. We simulated an external force pulling on *o*-MeOBuDPP or *m*-MeOBuDPP, which are alkylated PhDPP versions used later experimentally, as depicted in Fig. 3a. Our model structures show similar effects of steric hindrance in their equilibrium structures at zero force compared to BrPhDPP, whereby the methoxybutyl substituent in *o*-MeOBuDPP provides slightly less repulsion with the DPP core than the Br atom in *o*-BrPhDPP.

As the external force increases, the distance between pulling points and the central part of the molecule increases too. This leads to a deformation of the molecule, which reduces steric hindrance caused by the interaction of *ortho*-substituents at the phenyl ring and the DPP core (Fig. 3b, Supplementary Figs. 3 and 4). This effect is significantly stronger for *o*-DPP due to its larger *ortho*-substituent and larger initial dihedral angle. In both isomers, the dihedral angle decreases until the molecule eventually reaches a nearly coplanar structure (Fig. 3b). Thus, the primary cause for planarization is not the direct force acting on the DA spring, rather the tendency of the DA system to increase π orbital overlap and thereby reduce its energy, which becomes increasingly possible as a sterical hindrance is relieved. Therefore, the independent rocking of the central DPP core is restricted (cf. Fig. 2b). The two weakest Me-O bonds break spontaneously at larger forces of around 6 nN.

The force-induced planarization has a clear effect on the optical properties as revealed by time-dependent DFT (TDDFT) calculations in Fig. 3c and d. The absorption maximum $\lambda_{max,abs}$

shows a clear bathochromic shift with increasing external force. This effect is strongest in *o*-MeOBuDPP which displays a large total absorption shift $\Delta\lambda_{max,abs}$ of 115 nm at 6 nN, whereas for the analogous *meta*-derivative $\Delta\lambda_{max,abs} = 50$ nm at 5.9 nN is obtained. Figure 3c reveals that the shift is caused both by the rotation of the phenyl rings, as well as by the deformation of PhDPP. There is a striking similarity in the CT peak positions between *ortho*-springs and *meta*-springs for pure rotation (without action of an external force). The deviation of these two modes at small dihedral angles is caused by steric repulsion between the Ph rings and DPP cores, which leads to an additional deformation and a stronger shift in *o*-PhDPP. In particular, for *m*-MeOBuDPP, deformation is the main source of the shift due to the initially smaller rotation. Accordingly, the possibility for relaxation in this mode under the action of the external force is decreased. Further analysis of the effect of deformation is presented in Supplementary Information (Supplementary Figs. 5–11), indicating that the elongation of C-C bonds in the DPP core has the strongest influence on the wavelength of the CT peak.

With a linear fit up to ~1 nN, which is expected to be a reasonable range for the forces present in this experiment[23], we predict shifts $\Delta\lambda_{max,abs}$ depending on the force *F* acting on the DA system of $\Delta\lambda_{max,abs}/F = 7 \pm 1$ nm nN$^{-1}$ and $21 \pm 1$ nm nN$^{-1}$ for *m*-MeOBuDPP and *o*-MeOBuDPP, respectively. These numbers are obtained from the rotamer with *trans*-up-down conformation. Other rotamers of *m/o*-MeOBuDPP (Supplementary Table 1 and Supplementary Fig. 12) are very similar in energy but show different behavior under external force. Assuming all four rotamers to be present with equal probability gives a slightly

lower averaged response of $\Delta\lambda_{max,abs}/F = 5 \pm 1$ nm nN$^{-1}$ and 17 $\pm$ 2 nm nN$^{-1}$ for *m*-MeOBuDPP and *o*-MeOBuDPP, respectively. An exchange of the methoxybutyl substituents by methyl groups (Supplementary Table 2 and Supplementary Figs. 13–17) and additionally considering all rotamers leads to enhanced shifts in the range of small forces: $\Delta\lambda_{max,abs}/F$ is about $12 \pm 1$ nm nN$^{-1}$ and $45 \pm 3$ nm nN$^{-1}$ for *m*-MeDPP and *o*-MeDPP, respectively. This result hints at the rigidity of the linker being important for force transduction. We will use these force-dependent shifts to estimate the molecular forces acting in the experiment below.

**Synthesis of a DPP based DA torsional spring.** In order to verify these predictions experimentally, we synthesized *ortho*-functionalized and *meta*-functionalized diphenyl DPPs and covalently incorporated them into a rigid, yet ductile, polymer matrix. Prior investigations with DA spring functionalized polydimethylsiloxane (PDMS) showed only very little mechanochromic response, which we ascribed to its soft nature (Supplementary Fig. 18). Despite the growing interest in DPP-based materials during the last two decades[32,35], *ortho*-substituted diphenyl DPPs have been rarely reported so far. One reason is certainly that coplanarity of DPP-containing conjugated materials is a common target[33]. Another factor may be related to the chemistry involved in *ortho*-substitution, which is significantly more challenging due to the increased steric hindrance. The few reported syntheses of *o*-substituted diphenyl DPP chromophores state poor yields of 2–7%[36,37]. Using the direct and efficient succinate synthesis route, the preparation of bis(*o*-bromophenyl) diketopyrrolopyrrole (*o*-BrPhDPPH) **2a** is displayed in Fig. 4a.

For the herein achieved, comparably high yield of 10% of **2a** (Supplementary Fig. 19), very slow addition of the succinate is crucial. However, with increasing reaction time debromination occurs, therefore the reaction was stopped after 15 h. The first evidence for the increased torsional angle of *o*-BrPhDPPH is its relatively high solubility. The lower ability of *o*-BrPhDPPH to aggregate compared to the *meta*-analog *m*-BrPhDPPH **2b** even allowed acquiring $^{13}C$ NMR spectra in good quality in the absence of solubilizing alkyl groups. After N-methylation of **2a**, most of the debrominated species could be removed by repeated recrystallization to obtain **3a** in 60% yield and with a purity of 97%.

In light of the results from theory highlighting beneficial rigid linkers (Supplementary Figs. 13–17), direct incorporation of **3a** into the polyphenylene P*mmp*P[23] via Suzuki polycondensation (SPC) appeared most straightforward. However, this synthetic approach failed (Fig. 4). Apparently, the sterically hindered 2-bromo group of **3a** reacts much slower compared to 5,5′-dibromo-2,2′-bis(hexyloxy)-1,1′-biphenyl (**M2**). For this reason, a butylene linker was introduced via Negishi coupling while maintaining the general *o*-substitution pattern, leading to **4a** in 14% yield[38]. To introduce the 4-bromophenyl unit required for SPC, ether synthesis with 4-bromophenol was straightforward and led to the monomer bis(*o*-(4-(4-bromophenoxy)butyl) phenyl)DPP (*o*-LBrPhDPP) **5a** in 38% yield.

Terpolymerization of 1 mol% of **5a** with 1,4-benzenediboronic acid bis(pinacol)ester (**M1**) and **M2** resulted in the statistically functionalized *o*-DPP-P*mmp*P (**6a**) with the number and weight average molecular weights from size exclusion chromatography (SEC) $M_{n,SEC} = 34$ kg mol$^{-1}$ and $M_{w,SEC} = 130$ kg mol$^{-1}$ after Soxhlet extraction (Fig. 4c). To provide reference material, the entire synthesis route was repeated with the same DPP chromophore (*m*-LBrPhDPP) **5b** having the point of attachment in *meta* position (Fig. 4b, d). The resulting *m*-DPP-P*mmp*P (**6b**) had $M_n = 33$ kg mol$^{-1}$ and $M_w = 84$ kg mol$^{-1}$. Differential scanning calorimetry (DSC) and thermogravimetric analysis of **6a** and **6b** furnished onset temperatures of 411 °C and glass transition

temperatures of 115 °C and 112 °C, respectively (Supplementary Figs. 20 and 21).

**Experimental investigation of the DA spring.** Experimental investigations of the mechano-optical properties were carried out on dumbbell-shaped films of the DPP-functionalized P*mmp*Ps **6a** and **6b**. The good agreement of UV-vis absorption spectra of monomers **5a, b** and polymers **6a, b** in solution (Supplementary Fig. 22) excludes significant interference between DA spring and polymer matrix and DA aggregation. During tensile testing, the UV-vis absorption band of **6a** shown in Fig. 5a flattens significantly with increasing elongation. The strong broadening at high strain suggests a force-induced broadening of the distribution of torsional angles.

A comparison of the stress-strain curve with the corresponding values of absorption maxima $\lambda_{max,abs}$ (Supplementary Fig. 23b) illustrates the relationship between tensile stress and absorption behavior. It can clearly be seen that for strains larger than 60%, the bathochromic shift $\Delta\lambda_{max,abs}$ is strongly linked to the stress-strain behavior and can therefore be described as mechanochromic response with a maximum $\Delta\lambda_{max,abs}$ ~8 nm. The delay of the mechanochromic response below 50% strain is due to the combination of the experimental setup and specimen shape (see Supplementary Figs. 23 and 24), which does not guarantee maximum force localization centered at the optical path. A similar analysis of *m*-DPP-P*mmp*P **6b** was not possible, because the shape of absorption spectra indicates several contributions (Fig. 5b). The higher resolution of the vibronic bands with increasing planarity of the π-system is already known from other DPP derivatives[39,40].

The rather moderate changes in absorption under stress motivated us to investigate photoluminescence (PL), which is generally more sensitive for the detection of a mechanochromic response[1,41]. Photoluminescence recorded from **6a** during tensile testing is depicted in Fig. 5c, which displays clearly stress-induced red-shifted spectra. The second advantage of photoluminescence compared to UV-vis absorption is the further reduced contribution of the polyphenylene matrix due to the high PL intensity of the DPP chromophore (Supplementary Fig. 25). The stress-dependent PL spectra illustrate the continuous bathochromic shift of the emission band $\Delta\lambda_{max,em}$ of up to 19 nm with increasing stress (Fig. 5e). We assume that the retardation of mechanochromic response, i.e., between the onsets of necking and rise of $\Delta\lambda_{max,em}$, is an artifact, produced by the migration of the necking region into the optical path of PL detection and thus caused by the DIN norm of the specimen used (see Supplementary Figs. 23 and 24). Strikingly, a qualitative correlation between $\Delta\lambda_{max,em}$, and stress is observed, i.e., a stronger increase in stress produces a stronger $\Delta\lambda_{max,em}$. On the other hand, the rather constant stress during the continuous yielding between 20 and 150% strain results in a plateau of $\Delta\lambda_{max,em}$. Together with a negative control experiment using low molecular weight *o*-DPP-P*mmp*P (Supplementary Fig. 26), this confirms that the torsion potential of the DPP chromophore and the mechanochromic response of the DA spring can indeed be modulated by macroscopic force. Such a mechanism is clearly different from two-state systems, in which an increasing force commonly increases intensity but does not alter the wavelength of emission[2]. Upon rupture of the specimen, force is released and the *o*-DPP spring quickly returns to equilibrium seen by the PL spectrum perfectly matching the original spectrum of the non-strained state (Fig. 5c). While it is expected that a torsional profile enables such reversibility upon deviation from equilibrium geometry, the fast reversibility is still remarkable as the plastically deformed, tough P*mmp*P matrix with its high $T_g$ of 115 °C has undergone strains

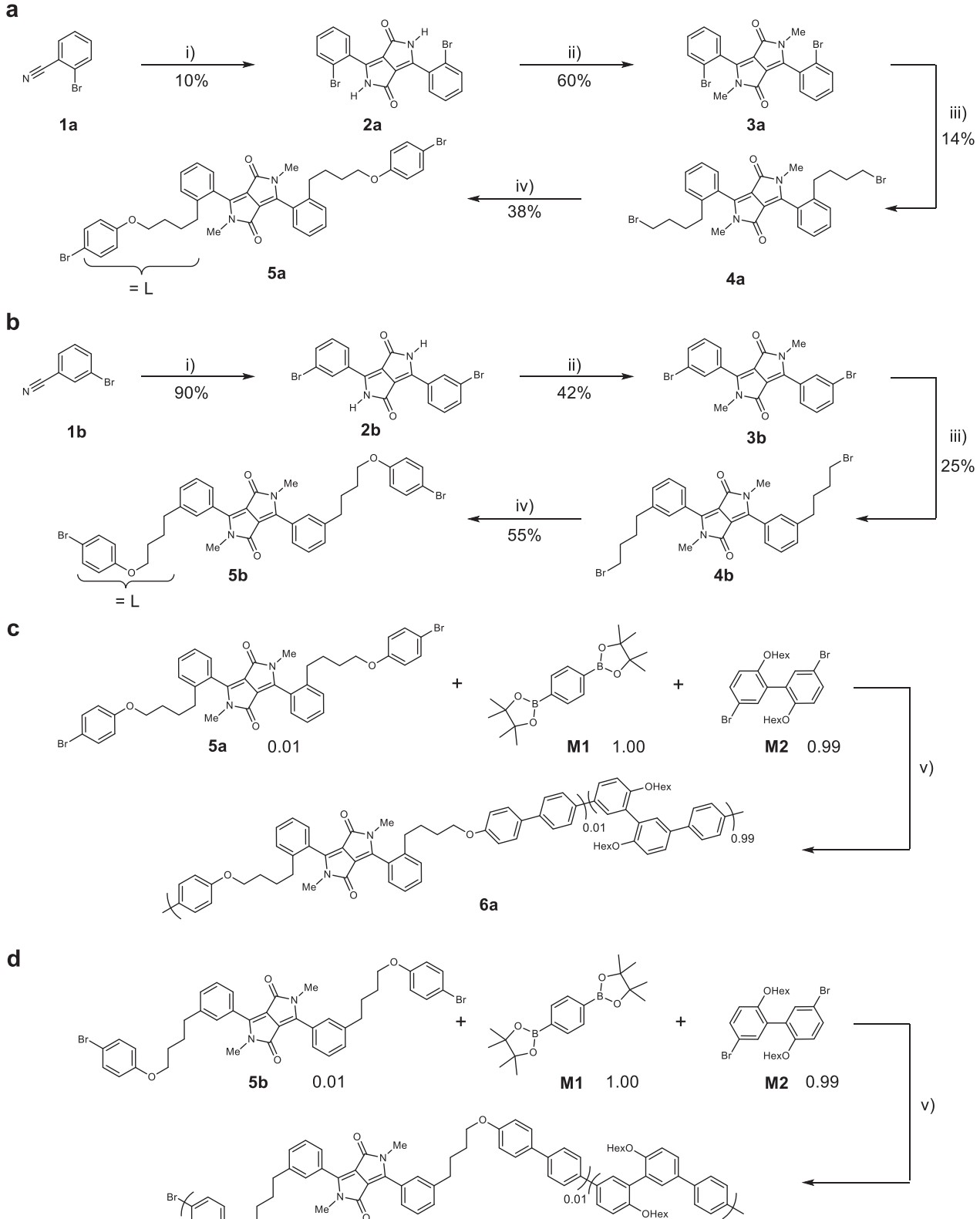

**Fig. 4 Synthesis route of the DA torsional springs. a, b** Synthesis of *o*-LBrPhDPP (**5a**) and *m*-LBrPhDPP (**5b**) with L = phenoxybutyl. **c, d** Incorporation of 5a and 5b into the polyphenylene matrix to *o*-DPP-P*mmp*P (**6a**) and *m*-DPP-P*mmp*P (**6b**). Reaction conditions: (i) diisopropylsuccinate, NaO*t*-Am, FeCl₃, *t*-AmOH, 100 °C, 15 h; (ii) Me-*p*-Tos, K₂CO₃, DMF, 120 °C, 3 h; (iii) 4-bromobutylzinc bromide (in situ), Pd₂(dba)₃, SPhos, DMAc, 80 °C, 2 h; (iv) 4-bromophenol, K₂CO₃, DMF, 50 °C, 16 h; (v) Pd₂(dba)₃, SPhos, Aliquat 336, toluene/2 M K₂CO₃ (aq), 70 °C, 24 h.

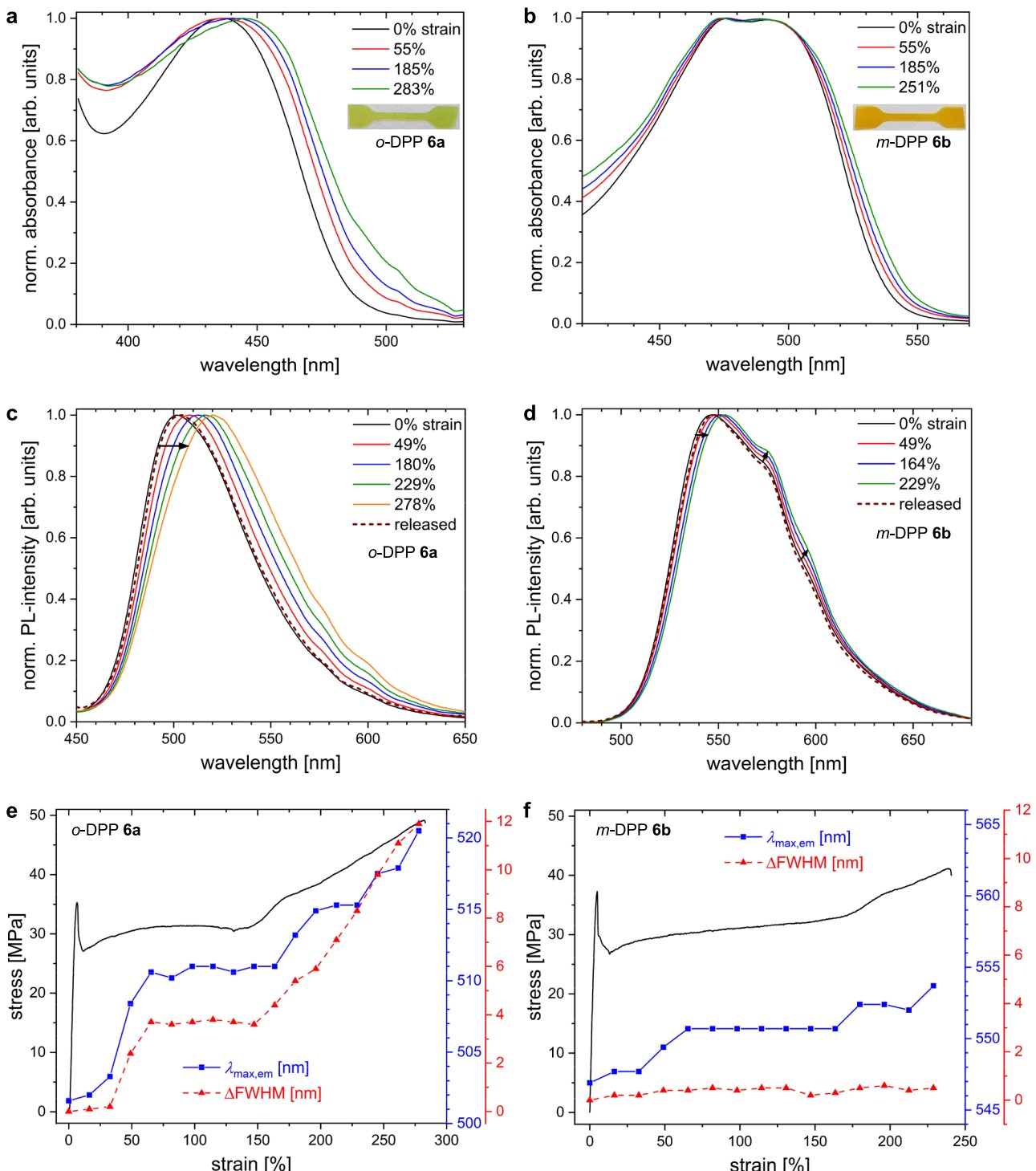

**Fig. 5 Combination of tensile testing and in-situ optical characterization of thin films.** Left column: *o*-DPP **6a**; right column: *m*-DPP **6b**. **a**, **b** UV-vis absorption spectra measured during/after stress-strain experiments and (**c**, **d**) the corresponding emission spectra. **e**, **f** Stress-strain curves (black) together with the detected emission maxima $\lambda_{max,em}$ (blue) and the changes in Full Width at Half Maximum of the emission bands ($\Delta$FWHM, red). Strain values as percentage elongation with respect to initial specimen length.

up to 300% at that time. Furthermore, Fig. 5c shows that the low energy onset of emission increases while the high-energy onset does not change much. This strain-induced broadening of the emission band is reflected by the increasing full width at half maximum (FWHM) in Fig. 5e. In addition to the overall increasing planarization, the distribution of DA torsional angles within the polymer becomes broader as not all springs are deformed in the same way.

Figure 5d reports the corresponding emission spectra of *m*-DPP **6b** also featuring a clear maximum. However, as for the absorption spectra, the more planar *m*-DPP derivative shows stronger vibronic features in the form of shoulders compared to *ortho*-DPP. The pronounced shoulder around 570 nm is also present in dilute solution (Supplementary Fig. 27), but its intensity is increased in solid-state of **6b**. The good agreement of the PL spectra of *m*-DPP monomer **5b** and polymer **6b**

(Supplementary Fig. 27) excludes an influence of P*mmp*P on the PL properties of the chromophore. Furthermore, different solvent polarities do not show a noteworthy effect on the emission behavior of the *m*-DPP unit (Supplementary Fig. 27). Under elongational force, the normalized spectra of **6b** in Fig. 5d show a bathochromic shift of $\lambda_{max,em}$, and growth of the low-energy band. Due to superposition, the increased intensity of the shoulder influences the position of $\lambda_{max,em}$. Interestingly, the FWHM changes only marginally (Fig. 5f, red dashed line). At strain at break (229%), *m*-DPP exhibits $\Delta\lambda_{max,em} = 7$ nm, which depicts a weaker mechanochromic response compared to *o*-DPP as expected from theory. Thus, the *m*-DPP spring clearly displays an overall weaker response. Due to different intensities of vibronic shoulders, the quantitative comparison of the mechanochromism between *o*-DPP and *m*-DPP on the basis of $\Delta\lambda_{max,em}$ values is not straightforward.

It is now interesting to compare the computed $\Delta\lambda_{max,abs}$ range with the much smaller experimentally observed values of $\Delta\lambda_{max,em}$, assuming a stress-independent Stokes shift. Usage of the theoretical fits from *m/o*-MeOBuDPP and the measured $\Delta\lambda_{max,em} = 7$ and 19 nm for *m*-DPP and *o*-DPP, respectively (Fig. 5c, f), results in estimated forces of 1.4 ± 0.6 and 1.1 ± 0.1 nN to act on the *m*-DPP and *o*-DPP springs, respectively (Fig. 3c). Thus, by combining theory and experiment it is possible to read out forces that act on single chains, which is not possible from experimental investigations alone.

**Reversibility of the mechanochromic response.** The reversibility of the *o*-DPP–based DA spring was further investigated in a back-and-forth stress-strain experiment, in which the stress was released three times by negative elongation (Fig. 6a). PL spectroscopy was measured during tensile testing and the time-dependent stress and $\Delta\lambda_{max,em}$ values were recorded. The images displayed in Fig. 6b show **6a** under UV-light (365 nm) at different times. At position 2 (50% strain), with the necking region of the sample having reached the light path of PL detection, **6a** showed a mechanochromic response $\Delta\lambda_{max,em}$ of 6 nm. The shift of $\lambda_{max,em}$ reversed immediately upon stress release (Fig. 6, pos. 3) and remained constant until further stretching of the sample. Similar behavior was observed during the following stress and release steps. Considering that tensile testing was performed 90 °C below the polymer's $T_g$, the quick reversibility of the DA spring is remarkable. Furthermore, despite the increasing deformation of the polymer matrix, the mechanochromic response $\Delta\lambda_{max,em}$ remains strongly dependent on the applied tensile stress until rupture of the specimen. At 263%

strain (Fig. 6b, pos. 4), the mechanically induced shift of 16 nm is clearly visible by the naked eye.

## Discussion

We have designed, simulated, and synthesized a mechanochromic donor-acceptor (DA) torsional spring based on *ortho*-substituted diphenyldiketopyrrolopyrrole (*o*-DPP), in which a stress-induced decrease of the dihedral angle between D and A caused a bathochromic shift of absorption and emission wavelength during tensile testing. Incorporation of the DA spring into an amorphous, high $T_g$, and tough polyphenylene matrix was key to experimentally observe spectral shifts. The theoretical calculations were crucial to minimize synthetic efforts and enable a deep understanding of the molecular properties and mechanistic details of the DA spring. The combination of simulations and experimentally determined spectral shifts allowed us to extract the effective microscopic forces that experimentally acted on a single DA spring in a quantitative manner. The maximum average force acting on the chains was estimated to be ~1.1 nN based on theoretical calibration. Thus, the tough polyphenylene matrix did not only enable deformation at room temperature but also provided high rigidity for transduction of sufficiently large forces. These estimated forces for a single chain are difficult to reconcile with macroscopically measured stress values of ~50 MPa. Establishing quantitative correlations between forces on single chains and macroscopic stress remains an open question and is the subject of ongoing investigations. As the mechanochromic response relies on a deflection of the equilibrium geometry of the DA spring, conformational and color changes were fully reversible on a short timescale even within the rigid polyphenylene matrix. The herein presented mechanochromic DA spring has several advantages over existing mechanophores, such as (i) fast response due to the missing barrier as present in two-state systems, (ii) reversibility for the same reason and (iii) continuous variation of wavelength with stress. These properties render the DA spring concept qualitatively different from other mechanochromic systems. In future studies, we will address the correlation of single-chain and bulk properties required for applications that involve real-time stress-sensing and mapping of force distributions.

## Methods

**Syntheses.** All details regarding the synthesis and characterization of small molecules and polymers are given in the Supplementary Information.

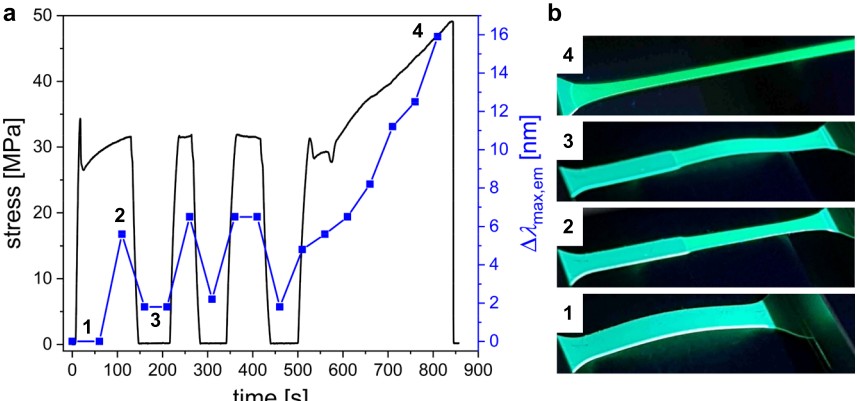

**Fig. 6 Reversibility of the mechanochromic response. a** Stress vs. time curve (black) of *o*-DPP **6a** during a stress-strain experiment with three relaxation breaks via negative elongation (5 mm min⁻¹ for both directions). The corresponding shifts $\Delta\lambda_{max,em}$ (blue) from PL-spectroscopy were correlated to the stress values by the time. **b** Images of the specimen under UV-light (365 nm) at the four indicated positions in the stress-time curve.

**Preparation of films**. 60 mg of DPP-P*mmp*P (**6a** or **6b**) were mixed with 120 mg pristine P*mmp*P and dissolved in 3 mL dichloromethane (DCM) followed by stirring for 30 min. The clear solution was filtered and cast into a petri dish (ø = 35 mm). The sample was covered with an inverted beaker and the solvent evaporated overnight in a fume hood at room temperature. The dumbbell-shaped specimens for mechanical testing were die-cut from the film using a standard punching tool (DIN 53504 type 3) having a width of 2 mm and an initial testing length between 18 and 20 mm. Typical thicknesses were between 100–130 μm. In order to observe full reversibility of the m-DPP spring, complete removal of DCM either by prolonged room temperature storage or the application of vacuum following film casting is necessary.

**Combination of tensile testing and optical characterization**. Tensile Testing was carried out on a Linkam TST-350 without closing the stress-strain chamber and by using a displacement ramp of 5 mm min$^{-1}$. The shown values of stress were calculated by considering the constant cross-section area of the specimen's mid-point at 0% strain. Absorption spectra were measured in transmission with a Flame-S UV-vis spectrometer from Ocean Optics. The spectra were smoothed via Savitzky-Golay filter considering 51 pts (**6a**) and 21 pts (**6b**). Excitation for photoluminescence (PL) spectra was carried out using a UV LED (Nichia NVSU233A UV SMD-LED, 365 nm, max. 1030 mW, operated with 900 mA) at a distance of 10 cm. The angle between exciting and detected light was 60 degrees. PL spectra were measured on the same spectrometer used for UV-vis spectroscopy and smoothed via Savitzky-Golay filter considering 21 pts. The data of optical properties and mechanical analysis was correlated via measuring time and include a maximum error in the strain of ±2%.

**Computational settings**. The electronic structure of the molecules was calculated using DFT as implemented in the GPAW package[42,43] where the exchange-correlation potential is modeled as devised by Perdew, Burke, and Ernzerhof[44]. Wave functions and electron density were described within the projector augmented wave method, where their smooth parts are represented on real-space grids with grid spacing 0.15 Å for the wave functions and 0.075 Å for the electron density. The grid was ensured to cover at least 4 Å around each atom. Optical properties were calculated by time-dependent DFT (TDDFT) in Casidas linear response formalism[45,46]. All transitions between Kohn-Sham orbitals within an energy range of 8 eV were considered in the linear response calculations, which were checked for convergence for excitations in the optical range (<3 eV) of interest here.

In order to explore the action and effects of external forces on the molecules, we used the method of constrained geometries to simulated external force (COGEF)[47–49]. This method restricts the distance $d$ between the two atoms where the external force is thought to act. Relaxation of all other degrees of freedom defines a potential $U(d)$ and the necessary external force $F(d)$ to satisfy the constraint. In the calculation, the force can be determined either from the derivative $\delta U \delta d$ or equivalently from the forces acting on the constrained atoms.

## Data availability

The authors declare that the data supporting the findings of this study are available within this article and its Supplementary Information files. The data generated during this study are available from the corresponding authors on reasonable request.

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

## Acknowledgements

The authors thank P. Godermajer for help with the synthesis of *m*-DPP and F. Kempe for the synthesis of M1 and M2. The authors acknowledge D. Adamczak for SEC, D. Stegerer for TGA, and A. Erhardt, L. Kassner, and M. Weber for MS measurements. M.R. thanks H. Komber for the discussion of NMR results. W.M. and M.W. are grateful for funding by the Deutsche Forschungsgemeinschaft (DFG, German Research Foundation) under Germany's Excellence Strategy–EXC-2193/1–390951807 and through Project WA 1687/11-1. M.S. greatly acknowledges funding from the DFG (SO 1213/11-1). W.M. and M.W. acknowledge computational resources by the state of Baden-Württemberg through bwHPC and the German Research Foundation (DFG) through grant no INST 39/963-1 FUGG (bwForClusters NEMO and JUSTUS2). The publication of this article was funded by the Chemnitz University of Technology.

## Author contributions

M.R. designed, synthesized, and characterized the materials of this work. W.M. performed DFT calculations. M.S. conceived the idea. M.S. and M.W. jointly supervised the work. M.R. wrote the first draft of the manuscript, and all authors were involved in discussing and revising it.

## Funding

## Competing interests

The authors declare no competing interests.
