## [Peer Review File · Nature Communications]

REVIEWER COMMENTS

Reviewer #1 (Remarks to the Author):

Raisch et al. report the targeted synthesis of a mechanochromic polymer material based on a donor-acceptor torsional spring. The authors design the mechanophore/chromophore with the help of quantum chemical calculations in a very elegant and exemplary manner. This illustrates the predictive power of modern quantum chemistry and deviates from the mainstream, where first experiments are performed and theory is used to put some seasoning to the results. The design was carefully transferred to synthesis. Polymer samples were subjected to mechanical tests using standardized sample geometry. The mechanochromic response was studied in absorption and emission. The results are very impressive and should be published, subject to minor revisions in the presentation.

Page 4, last paragraph: Three abbreviations for the mechanophore - PhDPPPh, PhDPP, DPP - are introduced, and it is not evident why they are all necessary or why the abbreviation changes with purpose.

Page 5, 1st paragraph results and discussion: First a hypsochromic shift is described, later a bathochromic shift, but nowhere the reference, i.e. the system that does not exhibit any shift, is described.

Page 6, Scheme 1: In the schematic spectra, one peak is shifting, the other one stays in place. What is the nature of the peak that stays in place?

Page 13, Figure 3: How is 100% strain defined? I suspect it is a standard in materials science, but it would be helpful to briefly explain it.

Reviewer #2 (Remarks to the Author):

In this article, Raisch et al. report a mechanochromic unit centered around a donor acceptor diketopyrrolopyrrole group. Its mechanochemical behavior is hypothesized to arise from a tension-induced planarization of the structure that ultimately leads to a shift in its CT band. Computation was used to link the change of dihedral angle (and from there the CT band) with the force applied. The mechanochromism is demonstrated in a polyphenylene polymer matrix. This probe presents the advantage of providing a continuous response to tension unlike most of the previously described systems which display a binary behavior. This represents a great advance in the field of mechanochromic molecules and I recommend it for publication in Nature Communication after revision.

The authors claim that the mechanochromic behavior finds its origin in the planarization of the structure, the extent of which is inferred from the change in dihedral angle. The calculated force accompanying the change in dihedral angle is very large (up to 6 nN from $\sim 60^\circ$ to $\sim 20^\circ$ for o-

MeOBuDPP) and suggests an important deformation of the molecular backbone.

Could the authors comment and evaluate how much of the change in optical properties is due to the rotation and how much is due to the deformation of the backbone? (by comparing the variation in CT in function of dihedral angle for MeDPP and MeODPP obtained from constrained rotation (like they did for BrDPP, Fig1d) or pulling).

Fig2: what is the origin of the change in wavelength below 1 nN for m-DPP when the dihedral angle barely changes in the force range?

Figure 3:

- what is the origin of the plateau between 50 to 100/150% strain?
- why vibronic bands are observed in meta adduct and not ortho?

Rotamers are reported in the SI for the ortho substituted derivatives. Have the authors investigated the influence of rotamerism on the mechanical behaviour?

The mechanochromism has been demonstrated in rigid polymer matrix. Is the effect still observed in a more elastic polymer?

Other comments:

Abstract could be clearer and structured following Nature recommendation.

Scheme 2: structures are rather small.

Line 378: 'The maximum force acting on one chain is estimated to be 0.9 nN.' This statement should be clarified because I don't think it's necessarily correct. Force values determined by COGEF are typically higher than forces measured at low loading rate (like in the tensile test). Chains might adopt an orientation in which the pulling is not as effective as the ideal case observed in the COGEF calculations so a higher force might be needed to achieve the same response.

SI:

Computation methods could be reported in more detail.

COGEF curves should be reported (not just the result of it).

No MS reported.

No hyphen in H-NMR

Reviewer #3 (Remarks to the Author):

This work suggests how tension can control the optical properties of a conjugated polymer, making it act as a "grayscale" rather than "on/off" mechanochromophore. The concept is similar to that published previously by others (<https://pubs.acs.org/doi/abs/10.1021/mz5003323>), which should be cited, although the prior work was not taken to bulk materials.

The empirical data seem sound, the molecular design is clever, and the calculations are critical. But I

think the core conclusions require additional justification before publication. In particular, I am concerned that the changes in spectroscopy are not due specifically to the force-induced stretching of the polymer backbone as suggested. Main concerns are:

1. The calculated average forces for the observed shifts are HUGE. If the average force per chain is ~ 1 nN, wouldn't the observed stress be in the GPa range?
2. I worry that some (most?) of the observed spectral change is due to chain packing forces that grow as chains are aligned. This is very hard to rule out, especially since you need to keep the attachment points in the same place to get the same alignment, and that rules out the obvious control experiments. But I think that even if it would not be conclusive, the authors should investigate low-MW controls (too small for entanglements). What does it take to get a spectroscopic shift in embedded small molecules? Because low-MW controls will not align in the same way, a negative result here is not definitive. But a positive result would implicate factors other than tension in the observed behavior. This could explain the difference in ortho and meta behavior.
3. Do the C- and S- rotational isomers have different spectra, and can any of the spectroscopy changes be related to shifting that equilibrium, rather than distorting planarity in the S-isomers?
4. The fact that some of the changes in absorption are not fully reversible is consistent especially with hypothesis I point (2), above. It is otherwise very difficult to explain the irreversibility on basis of chain tension alone.

REVIEWER COMMENTS

Reviewer #1 (Remarks to the Author):

Raisch et al. report the targeted synthesis of a mechanochromic polymer material based on a donor-acceptor torsional spring. The authors design the mechanophore/chromophore with the help of quantum chemical calculations in a very elegant and exemplary manner. This illustrates the predictive power of modern quantum chemistry and deviates from the mainstream, where first experiments are performed and theory is used to put some seasoning to the results. The design was carefully transferred to synthesis. Polymer samples were subjected to mechanical tests using standardized sample geometry. The mechanochromic response was studied in absorption and emission. The results are very impressive and should be published, subject to minor revisions in the presentation.

Page 4, last paragraph: Three abbreviations for the mechanophore - PhDPPPh, PhDPP, DPP - are introduced, and it is not evident why they are all necessary or why the abbreviation changes with purpose.

Response: Thank you for this suggestion. We have now defined the different compounds in the main paper. "Ortho-substituted phenyl DPP" is "o-DPP", diphenyl DPP is "PhDPP". If a specific molecule is meant, "o-DPP" is followed by the molecule number (e.g. **6a**). We believe this should be clearer now.

Page 5, 1st paragraph results and discussion: First a hypsochromic shift is described, later a bathochromic shift, but nowhere the reference, i.e. the system that does not exhibit any shift, is described.

Response: The hypsochromic shift arises from sterical hindrance and is mentioned in comparison to the fully coplanar molecule in which there is no sterical hindrance, and which therefore does not act as a spring (hypothetically). We believe the text is straightforward and clear.

Page 6, Scheme 1: In the schematic spectra, one peak is shifting, the other one stays in place. What is the nature of the peak that stays in place?

Response: Thank you for this hint. We fully agree with the Reviewer that the picture of the peak staying in place (π - π star transition) was misleading. Initially we suspected that the strongest effects are found for the CT band absorption, therefore the schematic spectra. In fact, experiment as well as theory show that this is not the case. We have modified the spectra accordingly.

Page 13, Figure 3: How is 100% strain defined? I suspect it is a standard in materials science, but it would be helpful to briefly explain it.

Response: We have added a short definition in the caption of Figure 5 (number changed)

Reviewer #2 (Remarks to the Author):

In this article, Raisch et al. report a mechanochromic unit centered around a donor acceptor diketopyrrolopyrrole group. Its mechanochemical behavior is hypothesized to arise from a tension-induced planarization of the structure that ultimately leads to a shift in its CT band. Computation was used to link the change of dihedral angle (and from there the CT band) with the force applied. The mechanochromism is demonstrated in a polyphenylene polymer matrix. This probe presents the

advantage of providing a continuous response to tension unlike most of the previously described systems which display a binary behavior. This represents a great advance in the field of mechanochromic molecules and I recommend it for publication in Nature Communication after revision.

The authors claim that the mechanochromic behavior finds its origin in the planarization of the structure, the extent of which is inferred from the change in dihedral angle. The calculated force accompanying the change in dihedral angle is very large (up to 6 nN from $\sim 60^\circ$ to $\sim 20^\circ$ for o-MeOBuDPP) and suggests an important deformation of the molecular backbone.

Could the authors comment and evaluate how much of the change in optical properties is due to the rotation and how much is due to the deformation of the backbone? (by comparing the variation in CT in function of dihedral angle for MeDPP and MeODPP obtained from constrained rotation (like they did for BrDPP, Fig1d) or pulling).

Response: We thank the Reviewer for this question. We have extensively analyzed the effect of the external force on the DPP molecules in the revised manuscript and have added a new figure as fig. 3c and the corresponding discussion to the main text. From fig 3c the contributions of deformation and rotation can be seen. More details can be found in the revised SI, where the action of force on various distinct C-C bond lengths has been considered.

Fig2: what is the origin of the change in wavelength below 1 nN for m-DPP when the dihedral angle barely changes in the force range?

Response: The much smaller change of the dihedral angle for m-DPP as compared to o-DPP is explained by the smaller dihedral angle in the initial, force free configuration. This can be clearly seen from the new Fig. 3c of the main text.

Figure 3:

- what is the origin of the plateau between 50 to 100/150% strain?

Response: The plateau (now fig. 5e,f) is due to the necking process, during which the stress only changes locally (\rightarrow yielding), the total stress remains more or less constant. The matrix polymer is an amorphous, high Tg thermoplast with outstanding toughness. We have explained this in an additional sentence in the manuscript.

- why vibronic bands are observed in meta adduct and not ortho?

Response: As possible reasons we have ruled out aggregation and interactions with the polyphenylene backbone by comparing the optical properties of monomer and polymer, partly in different solvents (DMF vs. CF). Stronger vibronic bands are commonly seen for chromophores with increased planarity, and thus are reported for many planar DPP derivatives in the literature. We have added two additional references, see refs 39, 40.

Rotamers are reported in the SI for the ortho substituted derivatives. Have the authors investigated the influence of rotamerism on the mechanical behaviour?

Response: We thank the Reviewer for this remark and have investigated the effect of rotamers in the revised manuscript. The rotamers considered are all of very similar energy, but indeed show diverse mechanical behavior under force as discussed in the new Supplementary Figs. 12 and 13. Assuming all rotamers to be present with equal probability leads to qualitatively similar fits for the

dependence between wavelength and force. We have discussed this effect in detail in SI and added information also to the main text.

The mechanochromism has been demonstrated in rigid polymer matrix. Is the effect still observed in a more elastic polymer?

Response: Thank you for this important point. We had some time ago already prepared DA spring-functionalized PMDS elastomers, which did not or only very weakly show mechanochromic response. We have added this data now into the SI, and have mentioned this in the main paper. The bottom line is that PDMS is too soft and does not allow transducing sufficient force to the DPP spring.

Other comments:

Abstract could be clearer and structured following Nature recommendation.

Response: Thank you, we have changed the abstract according to Nature recommendation.

Scheme 2: structures are rather small.

Response: We have changed the chemical structures according to Nature recommendations.

Line 378: 'The maximum force acting on one chain is estimated to be 0.9 nN.' This statement should be clarified because I don't think it's necessarily correct. Force values determined by COGEF are typically higher than forces measured at low loading rate (like in the tensile test). Chains might adopt an orientation in which the pulling is not as effective as the ideal case observed in the COGEF calculations so a higher force might be needed to achieve the same response.

Response: The Reviewer is correct in that the forces obtained from COGEF are largely overestimating the forces needed to break a bond in an experiment because the effect of temperature is not included. Finite temperature introduces also a dependence on loading rate as we have discussed in Ref 49. The situation is different for the case of the torsional springs, where temperature might broaden the distribution of deformations and dihedral angles, but there is no bond to break and thus no barrier to overcome. The Reviewer is also correct in that the force might not always point into the most effective direction. We will thus see an average over all conformations of the polymers relative to the externally applied force. The experimentally observed shift is the average of the optical response of all DPP springs. The maximum shift seen in the experiment should then correspond to the maximum average force as determined from the computational scaling factor.

SI:

Computation methods could be reported in more detail.

COGEF curves should be reported (not just the result of it).

Response: Despite that a complete description of the computational methods is given at the end of the main manuscript, we have added some more details to the revised SI also. We furthermore follow the advice of the Reviewer and show the computational COGEF curves in Supplementary Fig. 4 now.

No MS reported.

Response: We have added mass spectrometry data for the new compounds

No hyphen in H-NMR

Response: thank you, done.

Reviewer #3 (Remarks to the Author):

This work suggests how tension can control the optical properties of a conjugated polymer, making it act as a “grayscale” rather than “on/off” mechanochromophore. The concept is similar to that published previously by others (<https://pubs.acs.org/doi/abs/10.1021/mz5003323>), which should be cited, although the prior work was not taken to bulk materials.

Response: Thank you for this hint, the mentioned ref has been cited now (ref 30).

The empirical data seem sound, the molecular design is clever, and the calculations are critical. But I think the core conclusions require additional justification before publication. In particular, I am concerned that the changes in spectroscopy are not due specifically to the force-induced stretching of the polymer backbone as suggested. Main concerns are:

1. The calculated average forces for the observed shifts are HUGE. If the average force per chain is ~ 1 nN, wouldn't the observed stress be in the GPa range?

Response: This is a good point. In fact, the maximal forces of 1.1 ± 0.1 nN and 1.4 ± 0.1 nN observed here are not that large compared to the reference given by the Reviewer (<https://pubs.acs.org/doi/abs/10.1021/mz5003323>), where internal forces of up to 2.5 nN were reported. Reverting the observed maximal stress of ~ 50 MPa would require an average cross section area per molecule of $\sim 2000 \text{ \AA}^2$, while the molecules can be estimated to have $\sim 50 \text{ \AA}^2$. The effect of necking will certainly have an influence here via a reduction of the effective cross-section. Nevertheless, we do not have a full explanation for this discrepancy at present, but will pick up the aspect of forces on single chains and bulk stress in future studies. We have added: “In future studies we will address the correlation of single chain and bulk properties required for applications that involve real-time stress-sensing and mapping of force distributions.”

2. I worry that some (most?) of the observed spectral change is due to chain packing forces that grow as chains are aligned. This is very hard to rule out, especially since you need to keep the attachment points in the same place to get the same alignment, and that rules out the obvious control experiments. But I think that even if it would not be conclusive, the authors should investigate low-MW controls (too small for entanglements). What does it take to get a spectroscopic shift in embedded small molecules? Because low-MW controls will not align in the same way, a negative result here is not definitive. But a positive result would implicate factors other than tension in the observed behavior. This could explain the difference in ortho and meta behavior.

Response: Thank you for this important comment. We have prepared and used a low molecular DA spring functionalized *PmmpP* sample with $M_{n,SEC} = 9$ kg/mol, see Supplementary Fig. 26, and blended it into a high MW *PmmpP* matrix to ensure similar overall mechanical properties. The resulting, vanishingly small mechanochromic response supports the initially reported principle of the DA torsional spring. Entanglements of the polymer backbone seem to be crucial for the stress-induced bathochromic shift of the PL band of o-DPP. We have also blended the monomeric spring into *PmmpP* and could not observe any change in the optical spectra (not shown, this is probably what the reviewer refers to as “obvious control experiments”). Moreover, we have repeatedly measured the m-DPP spring and do now observe full reversibility. It appears that preparation and

handling of the film is crucial, although we cannot fully explain the origin of the initially observed partial reversibility/ irreversibility. We have now included the new data as figure 5d,f.

3. Do the C- and S- rotational isomers have different spectra, and can any of the spectroscopy changes be related to shifting that equilibrium, rather than distorting planarity in the S-isomers?

Response: We thank the Reviewer for this remark. We have investigated the effect of rotamers in detail in the revised SI and have adjusted the fits for the force-dependent shifts accordingly as already mentioned above. While numbers slightly change, the overall discussion and concept of the DA spring is the same.

4. The fact that some of the changes in absorption are not fully reversible is consistent especially with hypothesis I point (2), above. It is otherwise very difficult the irreversibility on basis of chain tension alone.

Response: See answer to point 2, reviewer 3.

REVIEWER COMMENTS

Reviewer #1 (Remarks to the Author):

The authors have addressed my concerns thoroughly and resolved all discrepancies. I am happy to recommend publication of this fine work in its current form.

Reviewer #2 (Remarks to the Author):

My comments have been addressed and I recommend the manuscript for publication.

Reviewer #3 (Remarks to the Author):

The authors have done a nice job of responding to concerns raised in the initial review. The paper is much stronger, although I still have some lingering issues, perhaps mostly with wording.

I question the use of absolute phrasing such as "The mechanically induced deflection from equilibrium geometry of the DA spring is theoretically predicted and fully confirmed by experiments" in the abstract. The authors simultaneously state that the experiments confirm the theory, and also that theory allows them to estimate the forces in the matrix. Both of these cannot be true. Either the force-spectra are known from theory alone, and used to estimate the forces in the matrix, or the forces in the matrix are known, and used to confirm the theory. I think more careful language is appropriate throughout, and the use of wording such as "using the theoretical calibration, we estimate that..."

The actual force estimates still seem too high, as noted in the author response letter. I think this should be emphasized more in the manuscript. I encourage the authors to emphasize the qualitative behavior, and the fact that the relative behavior observed in materials of the two probes is quite similar to the relative effect of force computed on their spectra. This is a fabulous result.

The authors claim in the discussion that "The maximum force acting on one chain was estimated to be ~1.1 nN." but this is not true. The maximum *average* force acting on a chain was estimated to be 1.1 - 1.4 nN. And, again, this is maybe a factor of ~40X higher than expected. This really does concern me. It would be helpful if the authors could similarly embed a two-state force probe, such as a spiropyran or two, that have been calibrated computationally into the same materials to show that the behavior is consistent.

While the reversibility reported here is encouraging, it is also difficult for me to "unsee" the results presented in the earlier paper. The language about reversibility in the current paper therefore feels misleading. Some mention/explanation of both behaviors and the difference in the systems is, I think, appropriate. Otherwise, one wonders why only the result that aligns with the authors' story is presented. It does have a bit of a feel of interpretation being guided by expectation.

It does look like there is a slight shift in the PDMS samples (ESI). If the authors plot wavelength shift vs. stress for PDMS and the PE materials, do they appear to fall on the same line?

Reviewer #3 (Remarks to the Author):

The authors have done a nice job of responding to concerns raised in the initial review. The paper is much stronger, although I still have some lingering issues, perhaps mostly with wording.

I question the use of absolute phrasing such as "The mechanically induced deflection from equilibrium geometry of the DA spring is theoretically predicted and fully confirmed by experiments" in the abstract. The authors simultaneously state that the experiments confirm the theory, and also that theory allows them to estimate the forces in the matrix. Both of these cannot be true. Either the force-spectra are known from theory alone, and used to estimate the forces in the matrix, or the forces in the matrix are known, and used to confirm the theory. I think more careful language is appropriate throughout, and the use of wording such as "using the theoretical calibration, we estimate that..."

Response: The reviewer is correct in that there is no direct way to obtain the microscopic forces from experiment. We have changed this sentence in the abstract accordingly. Regarding further related changes see below.

The actual force estimates still seem too high, as noted in the author response letter. I think this should be emphasized more in the manuscript. I encourage the authors to emphasize the qualitative behavior, and the fact that the relative behavior observed in materials of the two probes is quite similar to the relative effect of force computed on their spectra. This is a fabulous result.

Response: We followed the advice by the Reviewer and have rephrased in the discussion section: "These estimated forces for a single chain are difficult to reconcile with macroscopically measured stress values of ~ 50 MPa. Establishing quantitative correlations between forces on single chains and macroscopic stress remains an open question and is the subject of ongoing investigations."

The authors claim in the discussion that "The maximum force acting on one chain was estimated to be ~ 1.1 nN." but this is not true. The maximum *average* force acting on a chain was estimated to be 1.1 - 1.4 nN. And, again, this is maybe a factor of $\sim 40X$ higher than expected. This really does concern me. It would be helpful if the authors could similarly embed a two-state force probe, such as a spiropyran or two, that have been calibrated computationally into the same materials to show that the behavior is consistent.

Response: The reviewer is correct, what is measured should be the maximal average force and we have made this explicit in the manuscript now. Regarding absolute values, see also last comment. Indeed, copolymerizing SP and the DA spring simultaneously is an interesting idea, where the spring could measure the force at which SP isomerizes to MC, and this would give a decent cross check. This would address the open question of the correlation of averaged forces in the DA spring with the rupture force (which is temperature- and loading rate-dependent...). However, we believe the analytical characterization of such specimen with two different chromophores is not at all trivial and is a study on its own. We also think that significantly more theoretical investigations would be required in parallel, which makes this a challenging endeavor.

While the reversibility reported here is encouraging, it is also difficult for me to "unsee" the results presented in the earlier paper. The language about reversibility in the current paper therefore feels misleading. Some mention/explanation of both behaviors and the difference in the systems is, I think, appropriate. Otherwise, one wonders why only the result that aligns with the authors' story is presented. It does have a bit of a feel of interpretation being guided by expectation.

Response: We understand the point of the referee. For the o-DPP spring, we never observed such discrepancy. Only for the meta-spring, we found that the first experiment gave the data we had presented in the first version of the paper (partial reversibility, blue data points). During the revision stage and upon repetition of the experiment using samples of slightly different treatments, we found that the starting wavelength of the PL maximum changed (see below) as well as the wavelength after force release (last data points). Sample preparation differed by storage time at RT, further thermal annealing of the application of vacuum. We explain this by residual solvent (DCM) that may have unintentionally been trapped in specimens used for the first experiment, which is removed by either time, temperature or vacuum. We believe that reversibility of the meta spring can be claimed from the below data (green data points are used in the present version of the manuscript), and we also believe that it is legitimate to discard the first experiment. We have also added a sentence in the methods section: “In order to observe full reversibility of the m-DPP spring, complete removal of DCM either by prolonged room temperature storage or the application of vacuum following film casting is necessary.”

It does look like there is a slight shift in the PDMS samples (ESI). If the authors plot wavelength shift vs. stress for PDMS and the PE materials, do they appear to fall on the same line?

Response: This is correct, we have extended SI Fig 18. The data points do not fall onto the same line. Considering the very different nature of elastomeric PDMS and thermoplastic and ductile *PmpP*, this is probably not surprising.

REVIEWERS' COMMENTS

Reviewer #3 (Remarks to the Author):

I accept the revisions made by the authors.